# Real-World Data on Cabozantinib in Advanced Osteosarcoma and Ewing Sarcoma Patients: A Study from the Hellenic Group of Sarcoma and Rare Cancers

**DOI:** 10.3390/jcm12031119

**Published:** 2023-01-31

**Authors:** Stefania Kokkali, Anastasios Kyriazoglou, Elpida Mangou, Panagiota Economopoulou, Michail Panousieris, Amanda Psyrri, Alexandros Ardavanis, Nikolaos Vassos, Ioannis Boukovinas

**Affiliations:** 1Department of Medical Oncology, Saint-Savvas Anticancer Hospital, 11522 Athens, Greece; 2Medical Oncology Unit, Department of Internal Medicine, Hippocratio General Hospital, National and Kapodistrian University of Athens, 11527 Athens, Greece; 3Medical Oncology Unit, Department of Internal Medicine, Attikon University Hospital, National and Kapodistrian University of Athens, 12462 Athens, Greece; 4Medical Faculty Mannheim, University Medical Center Mannheim, University of Heidelberg, 68167 Mannheim, Germany; 5Department of Medical Oncology, Bioclinic Hospital, 54622 Thessaloniki, Greece

**Keywords:** osteosarcoma, Ewing sarcoma, bone sarcoma, cabozantinib, tyrosine kinase inhibitor

## Abstract

Advanced osteosarcomas (OSs) and Ewing sarcomas (ESs) tend to have poor prognosis with limited therapeutic options beyond first-line therapy. Aberrant angiogenesis and MET signaling play an important role in preclinical models. The anti-angiogenic drug cabozantinib was tested in a phase 2 trial of advanced OS and ES and was associated with clinical benefits. We retrospectively analyzed the off-label use of cabozantinib in adult patients with advanced OS and ES/primitive neuroectodermal tumors (PNETs) in three centers of the Hellenic Group of Sarcoma and Rare Cancers (HGSRC). Between April 2019 and January 2022, 16 patients started taking 60 mg of cabozantinib for advanced bone sarcoma or PNET. Median age at cabozantinib initiation was 31 years (17–83). All patients had received peri-operative chemotherapy for primary sarcoma and between 0 and 4 lines of treatment (median; 2.5) for advanced disease. The most common adverse effects included fatigue, anorexia, hypertransaminasemia, weight loss, and diarrhea. One toxic death was noted (cerebral hemorrhage). Dose reduction to 40 mg was required in 31.3% of the patients. No objective response was noted, and 9/16 patients exhibited stable disease outcomes. Progression-free survival varied from 1 to 8 (median; 5) months. Our study demonstrates that cabozantinib has antitumor activity in this population. In the real-life setting, we observed similar adverse events as in the CABONE study and in other neoplasms.

## 1. Introduction

Primary bone sarcomas are rare tumors that constitute a heterogeneous group of malignant mesenchymal neoplasms. In adults, osteosarcoma (OS) accounts for approximately 30% and Ewing sarcoma (ES) for approximately 10% of cases [1], whereas in children and adolescents they are the most common primary bone sarcomas [2]. The management of localized, conventional, high-grade OS and ES includes a complete surgical resection of the tumor combined with perioperative chemotherapy, leading to a cure rate of approximately 60%.

On the contrary, advanced OS and ES have a poor prognosis and show low response rates to the available chemotherapeutic agents used beyond first-line therapy. For high-grade OS, doxorubicin and cisplatin +/− methotrexate are used as the initial chemotherapy regimen [3]. Further lines of treatment include ifosfamide/etoposide [4], cyclophosphamide/topotecan [5], and docetaxel/gemcitabine [6]. For ES, alternating vincristine/doxorubicin/cyclophosphamide and ifosfamide/etoposide is usually administered in the perioperative and first-line setting [7]. Other systemic therapies include vincristine/ifosfamide/doxorubicin/etoposide [8], cyclophosphamide/topotecan [9], and temozolomide/irinotecan [10].

Aberrant angiogenesis and MET signaling have been shown to play a key role in preclinical models of ES and OS [11,12,13,14]. The anti-angiogenic drugs lenvatinib [15], regorafenib [16,17], and sorafenib [18] have been evaluated in small phase 2 trials, including 30–40 patients, in advanced osteosarcomas as ≥2nd-line treatment. They led to a median progression-free survival (PFS) rate of approximately 3–4 months. Regorafenib, in particular, is the only drug that was compared to the placebo, and was observed to confer a statistically significant PFS advantage for OS [17].

Cabozantinib is an oral multiple tyrosine kinase receptor inhibitor, targeting vascular endothelial growth factor receptor 2 (VEGFR2), and c-MET and RET protooncogenes as well. At present, it is approved for renal cell, medullary thyroid, and hepatocellular carcinomas. It was tested in advanced ES and OS in the CABONE study, a multicenter single-arm, two-stage, phase 2 trial and led to promising results, as recently reported by the French Sarcoma Group [19]. Our study represents a retrospective analysis of cabozantinib use in advanced bone sarcomas, which was performed in the centers of the Hellenic Group of Sarcoma and Rare Cancers (HGSRC) in order to register our real-world experience of the off-label use of this drug in this rare group of patients.

## 2. Materials and Methods

We performed a retrospective study in three high-volume sarcoma centers of the HGSRC. Patients diagnosed with OS or ES, who received cabozantinib for advanced disease (local recurrence and/or metastases), were included in the study. A patient with PNET was also included. Histological diagnosis was confirmed by a tissue pathological examination during the initial biopsy or/and surgery of the primary tumor. Cabozantinib was initiated at 60 mg/day and was continued until disease progression or unacceptable toxicity. A dose reduction to 40 mg/day was decided by the treating physician, due to adverse events.

We analyzed the clinical and histopathological data, as well as the treatment regimens and outcomes. The data were retrieved from medical records. The variables of interest were demographics, histological type, localization of the primary tumor and recurrent disease, perioperative chemotherapy protocol, age at cabozantinib initiation, adverse reactions to cabozantinib, prior therapies for systemic disease, and outcomes with cabozantinib (response according to RECIST v.1.1 criteria, [20]). The patients were monitored for adverse events at every follow-up assessment and whenever clinically indicated. Adverse events were graded according to the National Cancer Institute Common Terminology Criteria for Adverse Events version 4.0. The objective response rate (ORR) was defined as the percentage of patients who either had a complete or partial response. (Imaging was, in general, performed every three months).

Continuous variables were summarized using descriptive statistics, including median and range values. Categorical variables were summarized using descriptive statistics, including counts and percentages. The Kaplan–Meier method was used for survival analysis and correlations between survival outcomes, and potential prognostic features were analyzed using the log-rank test. Overall survival was defined as the time between cabozantinib initiation and death of any cause. Patients who were still alive were censored at the last follow-up date. PFS was defined as the time between cabozantinib initiation and disease progression or death by any cause. Due to the small sample size, we did not perform univariate and multivariate analyses. Statistical analyses were computed using SPSS 28 (IBM Corp., Armonk, NY, USA). Significance was defined at *p* < 0.05.

## 3. Results

### 3.1. Patients’ Characteristics

We retrospectively collected data from 16 patients that were initiated on cabozantinib treatment between April 2019 and January 2022, with a male-to-female ratio of 13:3.

Table 1 summarizes the clinicopathological characteristics of patients included in the study. The median age at cabozantinib initiation was 30 years (range; 15–76). Eleven cases of OS (of which one patient presented radiation-induced OS of the mandible), four cases of ES, and one case of primitive neuroectodermal tumor (PNET) were included. The primary tumor was located in the extremities (*n* = 8, 50.0%), spine (*n* = 4, 25.0%), pelvis (*n* = 2, 12.5%), mandible (*n* = 1, 6.3%), and adrenal gland (*n* = 1, 6.3%). At cabozantinib initiation, metastatic disease was present in the majority of patients. Nine patients presented with distant metastases only (56.3%), three patients with both distant metastases and local recurrence (18.8%), and two patients with local recurrence only (12.5%). For the remaining two patients, information on recurrent/metastatic disease localization was not available.

All primary tumors were surgically removed, and perioperative chemotherapy was administered to all patients prior to cabozantinib initiation. The most common perioperative regimen was PAM (cisplatin, doxorubicin, and high-dose methotrexate) for OS and VDC/IE (vincristin/doxorubicin/cyclophosphamide—ifosfamide/etoposide) for ES. Patients with localized OS, who were <40 years old at diagnosis, also underwent adjuvant treatment with mifamurtide. Prior to cabozantinib treatment initiation, 75% of the patients received at least one chemotherapeutic regimen and 0–4 (median; 2.5) prior lines of therapy for advanced disease. Table 2 illustrates all the previous therapies. In the OS subgroup (*n* = 11), four patients (36.4%) received cabozantinib as first-line treatment, whereas the combination of ifosfamide/etoposide was the most common form of chemotherapy used as initial treatment for advanced disease (*n* = 3; 27.3%), followed by gemcitabine/docetaxel (*n* = 2; 18.2%). In the ES/PNET subgroup (*n* = 5), gemcitabine/docetaxel were the most frequently used agents in the first-line setting (*n* = 2, 40.0%). Different agents were administered as second-line treatment and beyond, including temozolomide/irinotecan (*n* = 4; 80.0%) and rechallenged with the perioperative regimen (*n* = 2; 40.0%) in the ES/PNET subgroup. Among the seven OS patients who received ≥2 lines of treatment for advanced disease prior to cabozantinib, gemcitabine/docetaxel (*n* = 2, 28.6%) and ifosfamide/etoposide in combination with or without lenvatinib (*n* = 2, 28.6%) were the most prevalent regimens.

### 3.2. Safety

During cabozantinib treatment, twelve (75.0%) patients experienced at least one adverse event (AE) related to the drug. The most common AE was fatigue (*n* = 5, 31.3%), followed by anorexia and transaminasemia (*n* = 3, 18.8% each). All AEs are listed in Table 3 and almost all of them were grades 1–3. Two serious AEs (SAE), probably treatment-related, were noted; a 33-year-old female patient diagnosed with PNET of the adrenal gland and lung metastases presented with unilateral hemothorax 1.5 months post-cabozantinib initiation. Hemothorax could have occurred due to the necrosis of a subpleural lung nodule, although tumor shrinkage was not documented. She required hospitalization in the intensive care unit, after which the AE resolved. Cabozantinib was permanently discontinued. A 34-year-old male patient diagnosed with ES of the pelvis with lung and liver metastases presented with cerebral hemorrhage 1.5 months post-cabozantinib initiation, leading to death.

Five (31.3%) patients experienced recurrent intolerable grade 2 or 3 AE requiring a dose reduction to 40 mg. These AEs occurred during the first month of treatment in four out of five patients and in the third month in one patient. Toxic effects led us to study drug interruptions in two patients; an 83- and 63-year-old man interrupted cabozantinib treatment, both for three months, for fatigue.

### 3.3. Treatment Exposure and Outcomes

The median duration of cabozantinib treatment was 5 months (1.6–38.2). Of the 12 evaluable patients, 9 patients (6 OS and 3 ES) exhibited stable disease (SD) outcomes and another 3 progressive disease (PD) outcomes as the best response. No objective response was recorded. Among the four non-evaluable patients, three died of the disease before any response assessment was performed, and for another, one assessment was not available.

At the data cut-off date for the survival analysis, two patients were alive and presented progression-free results. Eight PD events were documented. The median PFS was 5 (95% CI: 2.3–7.6) months, whereas the 3- and 6-month PFS rates following treatment initiation were 68.8% and 37.5%, respectively (Figure 1). PFS was >6 months for six patients, including four OS and two ES patients. PFS was >12 months for three patients (18.8%), including two OS and one ES patient. In the OS subgroup (*n* = 11), the median PFS was 5 (95% CI: 0.0–10.7) months, whereas the 3- and 6-month PFS rates following treatment initiation were 63.6% and 36.4%, respectively. In the ES/PNET subgroup (*n* = 5), the median PFS was 5.7 (95% CI: 2.8–8.6) months, whereas the 3- and 6-month PFS rates following treatment initiation were 80.0% and 40.0%, respectively. In the subgroup of patients ≤40 years old, the median PFS was 4.4 (95% CI: 0.4–8.4) months.

At the time of the analysis, three patients were still alive (three on treatment and three having switched to another therapy due to PD). Nine patients died of the disease and one patient died from cerebral hemorrhage. Median overall survival from cabozantinib initiation was 9.3 (95% CI: 5.8–12.8) months, as demonstrated in Figure 2. Five patients (31.3%) survived >1 year and two patients (12.5%) more than 2 years (long survivors). In the OS subgroup, median overall survival was 10.0 (95% CI: 4.3–15.7) months, whereas in the ES/PNET subgroup 8.0 (95% CI: 0.2–15.8) months.

## 4. Discussion

Our study provides real-world data on the effectiveness and safety of the anti-angiogenic agent cabozantinib for advanced OS and ES cases, and further supports the results of the CABONE study [19]. Real-world data are valuable tools for performing a more thorough drug benefit–risk ratio evaluation, studying the use of the drug in a larger population, representing a real clinical setting. Adult bone sarcomas are extremely rare tumors, and evidence for the treatment of advanced-stage disease is limited to small single-arm retrospective studies of chemotherapeutic regimens. Recently, the first randomized data on the effectiveness of different chemotherapy combinations were reported for ES, favoring high-dose ifosfamide [21]. There is also some evidence on the effectiveness of tyrosine kinase inhibitors (TKIs), mainly for OS [15,16,17,18,19,22,23], with cabozantinib being the only drug showing benefits in ES [19].

Our study included adult patients between 17 and 83 years old, with a median age of 30 years, and thus the use of the drug in younger patients cannot be supported. However, given the higher prevalence of the disease in adolescents, most of the studies reporting data on anti-angiogenic drugs for advanced OS and ES include younger patients as well [15,19,22]. According to the CABONE study, patients younger than 16 years old received cabozantinib at a dose of 40 mg/m² [19]. Our results suggest that cabozantinib can be safely administered to patients > 40 years old, with a similar efficacy. Approximately 81% of our population is comprised of males, which is more than in other studies [15,16,17,18,19,23]. The vast majority of our patients were diagnosed with lung metastases, in line with all other studies.

The median line of prior therapies for advanced disease was 1.5, with cabozantinib administered as first-line treatment in four patients (25%). These four patients were treated with standard-of-care front-line therapy for early disease. Cabozantinib was opted in the first line of therapy, due to the early recurrence following the end of adjuvant chemotherapy (*n* = 2), old age (*n* = 1), or its safety profile (*n* = 1). In the CABONE study, a large proportion of patients were heavily pretreated, with a median number of two prior lines [19]. All anti-angiogenic agents were tested in the majority of patients after at least one prior line of treatment. Notably, in the phase 2 trial of lenvatinib for recurrent/metastatic OS, 4/31 patients received prior anti-VEGF therapy [15].

We also assessed tumor response to cabozantinib per RECIST and as per common clinical practice. We did not observe any objective response, whereas SD was noted in more than half of the patients. The objective response rate (ORR) of the different TKIs varied from 6.7% [15] to 26% [19]. The small sample size of our study, coupled with the observational design (different populations from the abovementioned randomized phase 2 studies), explain, at least partially, the absence of an objective response. The observed 56.3% disease control rate was very similar to regorafenib [17], higher than the one observed for sorafenib [18] and lower compared with the multi-target TKI anlotinib [23] and real-world data on the anti-VEGFR2 TKI apatinib [22], and almost identical to the one reported in the CABONE trial (46 pts with SD and PR out of 81 evaluable outcomes, 56,7%). Apart from the different properties of each agent (different targets of the kinase inhibitor), it is likely that variations in the duration of the observation period, the response assessment criteria used, as well as patient and disease characteristics may have contributed to these differences.

Treatment with cabozantinib resulted in a median PFS of 5.0 months (5 months for OS patients and 5.7 months for ES/PNET patients), which is higher than the 3- to 4-month range reported across some TKI clinical trials for OS [15,16,17,18], but comparable to the 5.2 months reported for anlotinib [23]. Despite the relatively small median values, regorafenib was shown to significantly prolong PFS compared to the placebo in both randomized trials [16,17]. The CABONE trial reported a median PFS of 6.2 months in the overall population, with a PFSR at 6 months of 33% for OS patients [19]. The PFSR at 6 months in the present study reached 36.4% for OS and 40% for ES/PNET patients, confirming drug activity. The longest PFS rate was observed in the retrospective study of apatinib, with a median value of 7.5 months [22].

Differences in PFS among studies could be due to several factors, such as patient and disease characteristics, including number of prior lines, timing of relapse, age of the patients, tumor burden, and histology. The different study design (prospective versus retrospective), different populations included (OS only versus both OS and ES/PNET), and the small numbers in all studies did not allow for a reliable comparison of the drugs across the studies. However, it should be noted that the median PFS with chemotherapy in relapsed OS is comparable to these results, if not shorter [24,25]. These direct comparisons with the results of the prospective trials should be interpreted with caution, since our retrospective analysis includes small numbers, non-rigid patient selections, and non-fixed disease assessment intervals.

A median overall survival rate of 9.3 months was reported, which compares well with the historical clinical trial data of pretreated patients with metastatic/recurrent OS (7–14 months) [16,17,18,19,22,23] and ES [26,27]. Interestingly, we observed two long-term survivors (>2 years since cabozantinib initiation), a young male with ES of the femur and lung metastases and a young female with OS of the tibia and lung metastases. While the long overall survival rate is correlated with a long PFS in the first patient, survival in the latter probably reflects the biological behavior of her disease and the effect of pulmonary metastasectomies, rather than the efficacy of cabozantinib treatment.

With respect to safety, treatment-related AEs observed in our study are consistent with the unwanted consequences of TKIs in OS trials [15,16,17,18,19,22,23]. Interestingly, the TKI lenvatinib has also been studied in combination with cytotoxic chemotherapy and the regimen was observed to be safe [28]. Overall, cabozantinib was well-tolerated, with no unexpected safety signals arising. However, a large proportion of our patients (75%) experienced at least one AE, in agreement with that reported in the CABONE trial (close to 100%). Dose reduction was more frequent in our population, compared to the CABONE trial (31% versus 21%). It should be noted that two patients experienced a probably treatment-related bleeding disorder leading to hospitalization. The first patient presented with hemothorax. Pneumothorax was observed in several clinical trials of TKIs in bone sarcomas [15,18,19] in a small proportion of patients (2–19%) and was associated with the presence of lung metastases. In the CABONE study, some cases of pneumothorax were reported in patients with pleural or sub-pleural lesions (both responding and non-responding to the drug), with a cavitation of lung metastases secondary to cabozantinib-induced tumor necrosis in some of them. In the study on lenvatinib [15], the presence of lung metastases and prior lung radiation therapy were observed to be risk factors for pneumothorax. Similarly, in the study on sorafenib [18], pneumothorax was observed in one patient and was attributed to bronchopleural fistula following a substantial reduction in a pleural metastasis. The second patient unfortunately experienced a cerebral hemorrhage with a fatal outcome. Hemorrhagic complications of anti-angiogenic drugs have been reported and can be life-threatening, requiring the close monitoring of patients [29]. Other extremely rare, potentially fatal complications of anti-angiogenic agents have been reported in the literature, such as colonic perforations with regorafenib [16].

We acknowledge that our study has some limitations, which are mainly attributable to its observational design, lack of central pathology review by an expert pathologist, missing data, and lack of internal controls. Even though physicians were encouraged to employ the same assessment technique and criteria over the study period, a response assessment was not centrally reviewed; hence, this may be influenced by observer bias. The non-comparative design of our retrospective, single-arm study is a major limitation that does not enable any definite conclusion for the drug’s efficacy to be formed. Finally, the limited sample that fits with the ultra-rare disease setting did not allow for an extensive analysis of the associations of treatment outcomes with patient and disease characteristics and may have impacted the precision of the estimation of the primary outcome measures. However, considering the rarity of the disease, the sample size, but also the fact that the patients were recruited from three oncology centers of the HGSRC, had a good semblance to the advanced adult bone sarcoma population in Greece.

## 5. Conclusions

Notwithstanding the favorable cure rates of early OS and ES/PNET and the chemosensitivity of the disease, 30–40% of patients will relapse. The prognosis of adult patients who progressed following standard multimodal therapy is dismal. Given the low response rates of multi-agent chemotherapy in relapsed/metastatic ES/PNET and OS patients, the paucity of evidence for effective treatment, and the high toxicity rates of these regimens, we consider cabozantinib a reasonable therapeutic option in patients who have either experienced an early relapse or disease progression following standard-of-care chemotherapy or were unsuited to receive front-line agents. Despite the non-comparative design and small sample size of the study, this small, real-life observational study further confirms the results of the CABONE phase 2 study. The favorable toxicity profile of cabozantinib, coupled with its administration on an outpatient basis, makes it a tentative agent. In conclusion, our retrospective study is the only one, apart from the CABONE trial, which supports the fact that cabozantinib provides a clinically meaningful benefit in terms of the clinical outcomes of adult patients with advanced OS or ES.

## Figures and Tables

**Figure 1 jcm-12-01119-f001:**
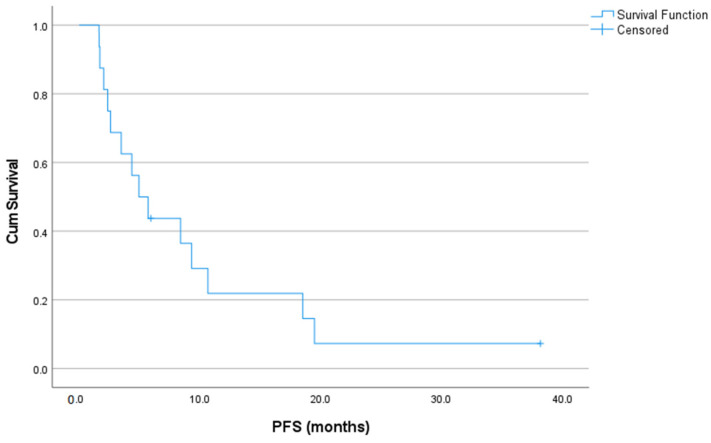
Kaplan–Meier curve of progression-free survival rates from cabozantinib initiation.

**Figure 2 jcm-12-01119-f002:**
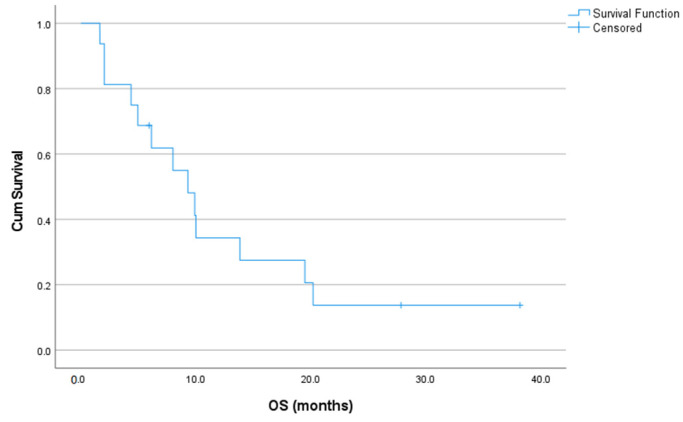
Kaplan–Meier curve of overall survival rates from cabozantinib initiation.

**Table 1 jcm-12-01119-t001:** Patient and disease characteristics at cabozantinib initiation.

Nr	Age	Sex	Diagnosis	Primary Location	Location of Recurrence
1	17	M	OS	radius	lung, bone, psoas
2	19	M	OS	tibia	local recurrence, lung
3	20	M	OS	humerus	lung
4	23	M	OS	femur	lung, kidneys, brain
5	33	F	PNET	adrenal gland	local recurrence, lung
6	63	M	OS	femur	lung
7	66	M	OS (RI)	mandible	local recurrence
8	83	M	OS	thoracic spine	local recurrence
9	34	M	ES	pelvis	local recurrence, lung, liver
10	47	M	OS	ilium	bone
11	18	M	ES	sacrum	lung
12	25	M	ES	thoracic spine	lung, pleura, muscle
13	29	F	OS	thoracic spine	NA
14	24	M	OS	femur	NA
15	31	M	ES	femur	lung
16	33	F	OS	tibia	lung

N: number, M: male, F: female, OS: osteosarcoma, ES: Ewing sarcoma, RI: radiation-induced, NA: not available.

**Table 2 jcm-12-01119-t002:** Treatment lines prior to cabozantinib initiation.

Nr	Prior Lines of Therapy	Perioperative Regimen	Prior Regimens for Advanced Disease
1	1	Cis/Doxo-->HD Metho	Ifo/VP
2	0	PAM, Mifamurtide	
3	0	PAM, Mifamurtide	
4	3	PAM, Mifamurtide	Ifo/VP, Gem/Doc, Cyclo/Topo
5	1	CAV	Ifo/VP
6	1	Cis/Doxo	Ifo/VP
7	1	Cis/Doxo	Gem/Doc
8	0	Cis/Doxo	
9	4	VDC/IE	rechallenge VDC/IE, Gem/Doc, Paz, Tem/Iri
10	3	PAM	Gem/Doc, rechallenge Gem/Doc, Ifo/VP
11	3	VDC/IE	Gem/Doc, Tem/Iri, rechallenge VDC/IE
12	2	VDC/IE	Gem/Doc, Tem/Iri
13	3	PAM, Mifamurtide	Carbo/VP, Ifo, Rego
14	2	Cis/Doxo/Ifo	Rego, Ifo/VP/Lenva
15	2	VDC/IE	Vin/Iri, Tem/Iri
16	0	PAM, Mifamurtide	

N: number, Cis/Doxo: cisplatin/doxorubicin, HD Metho: high-dose methotrexate, Ifo/VP: ifosfamide/etoposide, PAM: cisplatin/doxorubicin/high-dose methotrexate, Gem/Doc: gemcitabine/docetaxel, Cyclo/Topo: cyclophosphamide/topotecan, VDC/IE: vincristin, doxorubicin, cyclophosphamide/ifosfamide, etoposide, Paz: pazopanib, Tem/Iri: temozolomide/irinotecan, Carbo/VP: carboplatin/etoposide, Rego: regorafenib, Lenva: lenvatinib, Vin/Iri: vincristin/irinotecan.

**Table 3 jcm-12-01119-t003:** Cabozantinib-related adverse events and subsequent dose reductions.

Nr	Adverse Event	Dose Reduction
1		no
2	anorexia, fatigue, weight loss	40 mg
3	anorexia, fatigue, transaminasemia, weight loss	no
4	transaminasemia	no
5	hemothorax	no
6	dyspnea, fatigue	40 mg
7	fatigue	40 mg
8	fatigue, cytopenias	40 mg
9	cerebral hemorrhage	no
10	diarrhea	40 mg
11		no
12	dysthyroidism, oedema of lower extremities	no
13	hand–foot syndrome, hypertension, headache, nausea	no
14		no
15	anorexia, diarrhea, transaminasemia	no
16		no

## Data Availability

The data presented in the study are available upon reasonable request from the corresponding author.

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
