# Peer review of "Real-World Data on Cabozantinib in Advanced Osteosarcoma and Ewing Sarcoma Patients: A Study from the Hellenic Group of Sarcoma and Rare Cancers"

_jcm, 2023, doi:10.3390/jcm12031119_

Round 1

Reviewer 1 Report

Thank you for presenting this real-world data. Realizing it is a small cohort, osteosarcoma in older patients runs a different course than conventional osteosarcoma. While the authors touch upon this in the second paragraph of their discussion, an analysis of patients less than 40 years of age would be helpful. Furthermore the physiology of elder patients is quite different. 

Author Response

Response to Reviewer 1 Comments

Reviewer 1:

Thank you for presenting this real-world data. Realizing it is a small cohort, osteosarcoma in older patients runs a different course than conventional osteosarcoma. While the authors touch upon this in the second paragraph of their discussion, an analysis of patients less than 40 years of age would be helpful. Furthermore, the physiology of elder patients is quite different. 

Author’s response:

We thank you very much for your comments.

analyzed separately the population of patients ≤40 years old (n=12) in terms of PFS and overall survival and added the results of this analysis in the results section (treatment outcome). 

Furthermore, we totally agree that the physiology of elder patients is quite different; this is why we discussed this issue in the discussion section (2nd paragraph).

Reviewer 2 Report

Kokkali and colleagues present a retrospective analysis of relapsed osteosarcoma and Ewing sarcoma patients treated with cabozantinib between 2019 and 2022, reporting data on toxicity, response, and progression interval. 

-As a retrospective analysis on "real world" experience, I think the safety/tolerability/toxicity data is more valuable than the response and outcome data. I would consider moving the safety data to earlier in the results and expanding on this data.

-The text line 92 reports treatment lines from time of "advanced disease" whereas table 1 reports lines of treatment from diagnosis, please use one method of defining treatment lines and use consistently.

-Please include the median and range length of cabozantinib treatment for all patients.

-Please comment on any attempts at local control (surgery, radiation, etc) that occurred during cabozantinib treatment. 

-Please comment on any temporary drug holds that occurred during treatment, either for surgery or toxicity.

-The safety data analyzes adverse events. Please include in the methods a rigorous description of how adverse events were collected, graded, and attributed and what system was used with a citation (e.g. CTCAE?)

-Please comment on when in treatment the toxicities occurred that led to dose reductions.

-How did you define toxicities that were considered "related" to cabozantinib? Did you report toxicities of all grades that were "related" to drug?

-Table 3 is insufficient for the reporting of toxicities.  Need to include grades and give some indication of when in treatment the toxicity occurred and how long each lasted. There are also some toxicities listed that could incorporate several toxicities e.g. "cytopenias" and "transaminasemia". Recommend looking at other publications reporting toxicity data for guidance regarding formatting toxicity tables.

-Please include more detail on the SAE's reported. for the hemothorax: was this thought to be due to a lung nodule responding to treatment? For the cerebral hemorrhage leading to death, did the patient have an autopsy? can the authors think of any other contributing factors to cerebral hemorrhage other than cabozantinib?

-line 90 says some patients received mifamurtide in initial treatment-please include this in the perioperative regimens on table 2 

- in the discussion I recommend avoiding such direct comparisons of response and PFS with prospective clinical trials. Your retrospective analysis has small numbers, and non-rigid patient selection and disease assessment intervals which do not allow direct comparisons. 

Minor comments:

-line 70: "retrospectively" is misspelled

-line 86: does "operated" mean resected?

-line 89: I haven't heard the term "early OS" used before, i assume you mean younger patients?

Author Response

Response to Reviewer 2 Comments

Reviewer 2:

Kokkali and colleagues present a retrospective analysis of relapsed osteosarcoma and Ewing sarcoma patients treated with cabozantinib between 2019 and 2022, reporting data on toxicity, response, and progression interval. 

-As a retrospective analysis on "real world" experience, I think the safety/tolerability/toxicity data is more valuable than the response and outcome data. I would consider moving the safety data to earlier in the results and expanding on this data.

Author’s response:

Thank you very much for your great attention and your comments. We modified the order of the results subsections accordingly (red highlighted).

Reviewer 2:

-The text line 92 reports treatment lines from time of "advanced disease" whereas table 1 reports lines of treatment from diagnosis, please use one method of defining treatment lines and use consistently.

Author’s response:

Thank you for your remark. We did use the “prior lines of therapy” and change it accordingly in table 2 (red highlighted).

Reviewer 2:

-Please include the median and range length of cabozantinib treatment for all patients.

Author’s response:

We added this data in the “treatment exposure and outcome” paragraph (red highlighted).

Reviewer 2:

-Please comment on any attempts at local control (surgery, radiation, etc) that occurred during cabozantinib treatment. 

Author’s response:

Thank you for this interesting comment. There was no attempt to local control during cabozantinib treatment because all patients had already undertaken surgical therapy of the primary tumor, as stated in the patients’ characteristics paragraph (red highlighted).

Reviewer 2:

-Please comment on any temporary drug holds that occurred during treatment, either for surgery or toxicity.

Author’s response:

Thank you for your point. We added the relevant comment to the safety paragraph (red highlighted).

Reviewer 2:

-The safety data analyzes adverse events. Please include in the methods a rigorous description of how adverse events were collected, graded, and attributed and what system was used with a citation (e.g. CTCAE?)

Author’s response:

Thank you for your important remark. We added the appropriate description (red highlighted).

Reviewer 2:

-Please comment on when in treatment the toxicities occurred that led to dose reductions.

Author’s response:

Thank you for your comment. We added this information to the safety paragraph (red highlighted).

Reviewer 2:

-How did you define toxicities that were considered "related" to cabozantinib? Did you report toxicities of all grades that were "related" to drug?

Author’s response:

Toxicities of all grades that were related to drug are reported (red highlighted).

Reviewer 2:

-Table 3 is insufficient for the reporting of toxicities.  Need to include grades and give some indication of when in treatment the toxicity occurred and how long each lasted. There are also some toxicities listed that could incorporate several toxicities e.g. "cytopenias" and "transaminasemia". Recommend looking at other publications reporting toxicity data for guidance regarding formatting toxicity tables.

Author’s response:

Thank you for this interesting point. This detailed information is unfortunately not available for all patients. We can separate cytopenias to leucopenia, anemia and thrombopenia, but  they can also be considered and reported together, as in other publications.

Reviewer 2:

-Please include more detail on the SAE's reported. for the hemothorax: was this thought to be due to a lung nodule responding to treatment? For the cerebral hemorrhage leading to death, did the patient have an autopsy? can the authors think of any other contributing factors to cerebral hemorrhage other than cabozantinib?

Author’s response:

Thank you for your comment.

Additional information was added in the paragraph describing the SAE (hemothorax): “Hemothorax could be due to the necrosis of a subpleural lung nodule, although tumor shrinkage was not documented.”

In the CABONE study, some cases of pneumothorax were reported, all of which in patients with pleural or sub-pleural lesions (both responding and non-responding), and some of them also cavitation of lung metastases secondary to cabozantinib-induced tumour necrosis in some of them. Therefore, they consider them a risk factor for developing pneumothorax. In the study of lenvatinib in osteosarcoma (Gaspar et al), the presence of lung metastases an prior radiation therapy to the lungs were both considered as risk factor for pneumothorax. Similarly, in the study of sorafenib (Grignani et al.), pneumothorax was observed in one patient and was attributed to bronchopleural fistula following substantial reduction of a pleural metastasis. A summary of the above comments was added in the discussion session.

Regarding the cerebral hemorrhage, no other risk factor could be identified.

Reviewer 2:

-line 90 says some patients received mifamurtide in initial treatment-please include this in the perioperative regimens on table 2 

Author’s response:

That was modified accordingly in table 2 (red highlighted).

Reviewer 2:

- in the discussion I recommend avoiding such direct comparisons of response and PFS with prospective clinical trials. Your retrospective analysis has small numbers, and non-rigid patient selection and disease assessment intervals which do not allow direct comparisons. 

Author’s response:

Thank you for your remark. We agree with your concern but we think that we have to look at the results of the prospective trials and compare with ours, if we mention the limitations above. We added the following phrase: “These direct comparisons with the results of prospective trials should be interpreted with caution, though, since our retrospective analysis includes small numbers, non-rigid patient selection and not fixed disease assessment intervals. “   

Reviewer 2:

Minor comments:

-line 70: "retrospectively" is misspelled.

Author’s response:

Thank you very much, it was corrected.

Reviewer 2:

-line 86: does "operated" mean resected?

Author’s response:

Yes, it was replaced by “resected”.

Reviewer 2:

-line 89: I haven't heard the term "early OS" used before, i assume you mean younger patients?

Author’s response:

We mean early stage disease. To make it more clear, it was replaced by “Patients with localized OS”.

Round 2

Reviewer 2 Report

No further comments